# Morin Hydrate Reverses Cisplatin Resistance by Impairing PARP1/HMGB1-Dependent Autophagy in Hepatocellular Carcinoma

**DOI:** 10.3390/cancers11070986

**Published:** 2019-07-15

**Authors:** Mahendra Pal Singh, Hee Jun Cho, Jong-Tae Kim, Kyoung Eun Baek, Hee Gu Lee, Sun Chul Kang

**Affiliations:** 1Department of Biotechnology, College of Engineering, Daegu University, Gyeongsan, Gyeongbuk 38453, Korea; 2Immunotherapy Convergence Research Center, Korea Research Institute of Bioscience and Biotechnology, Daejeon 34141, Korea; 3Department of Biomolecular Science, University of Science and Technology (UST), Daejeon 34141, Korea

**Keywords:** hepatocellular carcinoma, cisplatin, chemoresistance, Morin hydrate, autophagy, PARP1

## Abstract

Chemoresistance is a major obstacle that limits the benefits of cisplatin-based chemotherapy in various cancers, including hepatocellular carcinoma. De-regulation of the poly(ADP-ribose) polymerase 1 (PARP1)/high-mobility group box 1 (HMGB1) signaling pathway has been proposed as an important mechanism involved in cisplatin-resistance. In this study, we investigated therapeutic potential of a natural flavonoid Morin hydrate against cisplatin-induced toxicity using the HepG2^DR^ multi-drug resistant cell line, which is derived from the HepG2 human hepatocellular carcinoma cell line. HepG2^DR^ cells were exposed to cisplatin and Morin hydrate alone or together after which autophagy and apoptotic signaling pathways were monitored by fluorometric assay and Western blot analysis. Xenograft mouse models were performed to confirm the in vitro effect of Morin hydrate. PARP1 was hyper activated in cisplatin-resistant HepG2^DR^ cells. Cisplatin-induced PARP1 activation resulted in chemoresistance via increased autophagy. The cisplatin/Morin hydrate combination was effective in the reversal of the HepG2^DR^ cell resistance via suppression of PARP1-mediated autophagy by regulating the HMGB1 and microtubule-associated protein 1A/1B light chain 3B (LC3) I/II. Moreover, PARP1 inhibition by 4-amino-1,8-naphthalimide or autophagy inhibition by a knockdown of the autophagy-related 5 (*ATG5*) gene resulted in sensitizing the HepG2^DR^ cells to cisplatin (CP) through activation of the c-Jun N-terminal kinase (JNK) pathway. In a mouse xenograft model, the treatment of cisplatin with Morin hydrate reversed the increased expression of PARP and HMGB1 and significantly suppressed tumor growth. These findings indicate dysregulated expression of PARP1 confers cisplatin-resistance via autophagy activation in HepG2^DR^ cells. Morin hydrate inhibits cisplatin-mediated autophagy induction, resulting in increased susceptibility of HepG2^DR^ cells to cisplatin cytotoxicity. The combination of Morin hydrate with cisplatin may be a promising therapeutic strategy to enhance the efficacy of conventional chemotherapeutic drugs.

## 1. Introduction

Hepatocellular carcinoma (HCC) accounts for approximately 80–90% of liver cancers and is the second leading cause of cancer deaths globally [1]. Currently, there is no definitive curative drug to treat HCC in the clinic, and chemotherapy is often used to control the progression of unresectable tumors [2]. However, available conventional chemotherapeutic agents show low response rates and poor survival benefits because of the development of multidrug resistance [3]. Although the mechanism of chemoresistance is not fully understood, recent evidence has revealed that stress-induced autophagy may contribute in part to the HCC chemoresistance [4]. Autophagy is a highly conserved process by which cytoplasmic components are sequestered in double-membrane vesicles, called autophagosomes, and delivered to lysosomes for proteolysis to maintain the energy homeostasis [5,6]. A number of stress-inducing factors promote the activation of autophagy, including nutrient deficiency, a high temperature or pH, and chemical treatment. The pro-survival role of autophagy contributes to cytoprotective events, which help cells develop chemoresistance, while its pro-death role results in cancer cell death [7]. A recent study has implicated poly(ADP-ribose) polymerase 1 (PARP1) in autophagy induction via DNA damage- or oxidative stress-mediated phenomena. However, the exact molecular mechanism of PARP1-mediated autophagy regulation in drug resistance is not fully understood [8]. Increasing evidence reveals different roles of autophagy in the induction of chemoresistance to antineoplastic therapies such as cisplatin (CP), doxorubicin, and numerous other drugs [9,10]. Therefore, development of novel chemotherapeutic agents or adjuvants that improve the therapeutic potential of conventional drugs by overcoming the drug resistance is a flourishing area of cancer research.

CP is a platinum-based conventional chemotherapeutic drug that is commonly used for treating various cancers, including HCC [11]. Although CP chemotherapy is a first-line treatment for several cancers, cancer cells develop intrinsic resistance to CP therapy [12,13]. Recent studies have reported that multiple factors, including insufficient DNA binding, increased detoxification and DNA repair, deregulated expression of transporters, and altered expression of genes involved in cell death pathways contribute to the development of cancer resistance to CP [14,15]. In particular, the role of PARP1 in CP-induced autophagy has been considered in the development of drug resistance [16].

Numerous studies have assessed natural compounds from traditional herbs in the treatment of cancer and revealed their cytotoxic effects and roles in the induction and regulation of autophagy [17,18]. Morin hydrate (MH; 3,5,7,2′,4-pentahydroxyflavone) is a natural bioflavonoid isolated from *Maclura pomifera* and found in various herbs, fruits, and wine [19]. The chemical structures of MH are characterized by the presence of two aromatic rings connected by a *c*-pyrone ring, in which polar hydroxyl groups are bound at various positions. These hydroxyl groups have been suggested to be responsible for the free radical-scavenging properties of MH and other naturally occurring bioflavonoids [20]. Because of its high reactivity against free radicals, MH shows potent immunomodulatory properties, promoting the maintenance of a healthy immune system [21]. Previous studies have also demonstrated that MH plays a role in regulating cell survival, promoting apoptosis, and sensitizing chemoresistant cells, and it shows other pharmacological properties, such as antioxidant and anti-inflammatory activities, both in vitro and in vivo [22,23].

In the present study, we investigated the therapeutic potential of a natural flavonoid Morin hydrate against cisplatin-induced toxicity using a multi-drug resistant hepatocellular carcinoma cell line. We demonstrate that dysregulated expression of PARP1 confers resistance to CP via autophagy activation in HepG2^DR^ cells and the combination of MH with CP mediates the autophagy turnover, resulting in increased susceptibility of HepG2^DR^ cells to CP cytotoxicity.

## 2. Results

### 2.1. MH Treatment Reverses an Acquired CP-Resistant Phenotype of HepG2^DR^ Cells

To examine the effect of MH on CP resistance, HepG2 human hepatocellular carcinoma cells were cultured with CP (2 µM). The surviving cells were then treated with progressively increasing CP concentrations during cell passages [24]. Parental HepG2 cells (wild-type, WT) and cisplatin-resistant HepG2 (HepG2^DR^) cells were treated with 20 μM of CP in the presence or absence of MH. Expectedly, HepG2^DR^ cells were more resistant to cytotoxicity against CP than were HepG2^WT^ cells. On the other hand, the treatment with MH reduced this resistance of HepG2^DR^ cells in a concentration dependent manner (Figure 1A). The characteristic features of CP resistance, including increased expression of excision repair cross-complementation group 1 (ERCC1) and ATP-binding cassette sub-family B member 1 (ABCB1), also known as P-glycoprotein 1 (P-gp), as well as increased ATPase activity [25,26], were evaluated after appropriate drug treatment in HepG2^DR^ cells. The expression level of ERCC1 was significantly increased in HepG2^DR^ cells in comparison to that of HepG2^WT^ cells, while MH exposure reduced this expression significantly in a concentration dependent manner (Figure 1B). The modulation of the P-gp activity was measured in a rhodamine-123 (cell-permeable dye) accumulation assay. The intensity of green fluorescence was significantly lower in HepG2^DR^ cells than that of HepG2^WT^ cells. However, the fluorescence intensity was significantly increased in CP-MH-treated HepG2^DR^ cells with respect to that of the control or untreated HepG2^DR^ cells (Figure 1C,D). Quantification of P-gp-associated ATPase activity showed that the basal ATPase activity in CP-resistant HepG2^DR^ cells was significantly increased than that in HepG2^WT^ cells and significantly decreased after CP-MH treatment (Figure 1E). Western blot analysis revealed significantly increased PARP1 expression in HepG2^DR^ cells in comparison to that of HepG2^WT^ cells, and a dose-dependent reduction of PARP1 in HepG2^DR^ cells after CP-MH treatment was observed. The LC3I/II level, a hallmark of autophagy induction, was also significantly increased in CP-treated HepG2^DR^ cells. Considering the direct and indirect association of LC3I/II with p62, we examined p62 expression after CP treatment of HepG2^DR^ cells and found that it was significantly reduced after 24 h. In contrast, the expression of LC3I/II was significantly reduced in HepG2^DR^ cells after CP-MH treatment compared with that in untreated cells. Conversely, the expression level of p62 was significantly increased in CP-MH-treated HepG2^DR^ cells (Figure 1F).

### 2.2. MH Suppresses PARP1-Mediated Autophagy Activation in HepG2^DR^ Cells

Previous studies have reported that CP-induced DNA damage leads to PARP1 activation, which plays a crucial role in the PARylation-mediated translocation of high-mobility group box 1 (HMGB1) from the nucleus to the cytoplasm. The cytoplasmic HMGB1 interlinks with the Beclin-1 protein to sustain autophagosome formation [27]. Hence, we evaluated the cytoplasmic and nuclear expression levels of PAR and HMGB1. The levels of cytosolic PAR and HMGB1 were significantly higher in CP-treated HepG2^DR^ cells than that in untreated HepG2^WT^ cells and were reduced in HepG2^DR^ cells after CP-MH treatment. However, the nuclear expression levels of PAR and HMGB1 were increased in CP-MH-treated cells with respect to those of untreated HepG2^DR^ cells (Figure 2A). Furthermore, immunocytochemistry analysis showed that the distribution of HMGB1 into strongly increased punctate structures was observed in the HepG2^DR^ cells but not in HepG2^WT^ cells and was significantly reduced after treatment with CP-MH. To confirm that cytoplasmic HMGB1 translocation depends on PARylation, we exposed HepG2^DR^ cells to 10 μM ANI (PARP1 inhibitor) alone or in combination with CP. Consequently, the numbers of punctate structures were significantly reduced and were similar to that in CP-MH-treated HepG2^DR^ cells and HepG2^WT^ cells (Figure 2B). These results suggest that MH regulates CP-induced PARP1 activation and HMGB1 translocation in HepG2^DR^ cells.

Based on the known association between CP-induced DNA damage and PARP1 hyper expression, we investigated the role of autophagy in CP-induced HepG2^DR^ cells through acridine orange (AO) staining. CP-treated HepG2^DR^ cells accumulated a bright red color in acidic vacuoles (due to the weak basic properties of AO). However, after 24 h treatment with CP-MH, the percentage of red-stained acidic vacuoles was significantly reduced in HepG2^DR^ cells compared with those in untreated HepG2^DR^ cells (Figure 3A). To confirm the accumulation of acidic vacuoles, we performed a Cyto-ID green fluorescent stain assay using DAPI as the background stain. Visual monitoring by fluorescence microscopy showed a bright green color of CP-treated HepG2^DR^ cells but not of HepG2^WT^ cells. However, this increased intensity of green fluorescence was significantly reduced in CP-MH-treated HepG2^DR^ cells (Figure 3B). We next evaluated the expression levels of major autophagy-associated proteins. Western blotting analysis showed that the expression levels of PI3KIII, ATG7, ATG5, and Beclin-1 were significantly increased in HepG2^DR^ cells and were reduced after CP-MH treatment in a concentration-dependent manner (Figure 3C).

Since HMGB1 was shown to play a crucial role in PARP1-regulated autophagy activation in CP-treated HepG2^DR^ cells, we next compared the effects of ANI and MH on CP-mediated autophagy activation. Western blot analysis showed significantly reduced expression of cytosolic PARP1, HMGB1, and LC3I/II and increased expression of p62 after CP-ANI and CP-MH treatments compared with those after the treatment of HepG2^DR^ cells with ANI alone or in untreated HepG2^WT^ cells (Figure 3D). To confirm that translocation of HMGB1 from the nucleus to cytoplasm is responsible for autophagy activation in HepG2^DR^ cells, we used ethyl pyruvate (EP) to block the cytosolic translocation of HMGB1. The cytosolic HMGB1 was reduced in CP-EP-treated HepG2^DR^ cells and was similar to that in CP-MH-treated cells; simultaneously, the expression level of LC3I/II was also significantly reduced. However, the expression of p62 significantly increased in HepG2^DR^ cells after CP-EP and CP-MH treatments, compared with that in EP-treated HepG2^DR^ cells and in untreated HepG2^WT^ cells (Figure 3E). These results suggest that CP-induced PARP1 expression and cytoplasmic translocation of HMGB1 appear to be the major causes of the development of CP resistance in HepG2^DR^ cells via autophagy activation. However, the regulatory effect of MH against PARP1 in CP-MH-treated HepG2^DR^ cells may increase the sensitivity to CP through suppression of HMGB1-mediated autophagy activation.

### 2.3. MH Enhances Apoptosis of HepG2^DR^ Cells against CP by Suppressing PARP1-Mediated Autophagy

Cytotoxicity of CP, MH, and CP-MH for HepG2^DR^ cells was evaluated using the MTT assay and LDH assay. Significantly decreased cell viability and increased LDH release was observed in CP-MH-treated HepG2^DR^ cells compared with those of HepG2^DR^ and HepG2^WT^ cells (Figure 4A,B). Moreover, CP-MH treatments increase the apoptotic morphology, such as cell shrinkage, and apoptotic cell death (Figure 4C,D). To assess the anti-proliferative effect of CP-MH, we performed the colony formation assay. The results showed that the number of colonies formed by CP-MH-treated HepG2^DR^ cells were significantly less than that formed by HepG2^DR^ and HepG2^WT^ cells (Figure 4E,F). To test whether PARP1/HMGB1-mediated autophagy activation reduces CP cytotoxicity, we treated cells with ANI and EP alone or in combination with CP, followed by the apoptosis assay. The results showed significantly reduced cell viability of CP-ANI- and CP-EP-treated HepG2^DR^ cells compared with that of CP-, ANI-, and EP-treated HepG2^DR^ cells and untreated HepG2^DR^/HepG2^WT^ cells; the results for the CP-ANI- and CP-EP-treated cells were consistent with those for CP-MH-treated cells (Figure 4G,H).

### 2.4. MH Promotes the Mitochondria-Mediated Intrinsic Pathway of CP-Induced Apoptosis in HepG2^DR^ Cells

Since oxidative stress stimulates the proliferation and drug resistance of cancer cells [12], we investigated the effects of CP-MH on oxidative stress in HepG2^DR^ cells. The level of intracellular reactive oxygen species (ROS) was significantly higher in control HepG2^DR^ cells than that in HepG2^WT^ cells, but was reduced by MH treatment (Figure 5A,B). Interestingly, MH treatment increased the level of Catalase, a major antioxidant enzyme, in CP-treated HepG2^DR^ cells (Figure 5C). To determine the effect of CP-MH on DNA damage-mediated apoptosis, we used the Hoechst 33358 stain. Increased pyknotic nuclei formation was observed in CP-MH-treated HepG2^DR^ cells compared with that in CP-treated HepG2^DR^ and untreated HepG2^WT^ cells (Figure 5D). Apoptosis is a process of programmed cell death that typically occurs via two pathways, known as intrinsic and extrinsic pathways. The intrinsic pathway of apoptosis involves mitochondria and is activated by a drop in the mitochondrial membrane potential (∆*Ψm*), release of cytochrome *c* (Cyt *c*) from the mitochondria to the cytoplasm, and activation of executioner caspases [28]. Therefore, we next examined the effect of CP-MH on ∆*Ψm* in HepG2^DR^ cells using the JC-1 stain. A significant loss in the red fluorescence intensity of JC-1 was observed after CP-MH treatment of HepG2^DR^ cells compared with that in untreated HepG2^DR^ cells and untreated HepG2^WT^ cells (Figure 5E,F).

Since MH regulates ∆*Ψm* in HepG2^DR^ cells, we next investigated the expression level of the Bcl-2 family of proteins. MH treatment increases the expression of Bax, a pro-apoptotic Bcl-2 family protein, and decreases the expression of Bcl-2, an anti-apoptotic Bcl-2 family protein. Moreover, the cleavage of caspase-9 and caspase-3 and the release of Cyt *c* were higher in CP-MH-treated HepG2^DR^ cells than those in CP-treated HepG2^DR^ and untreated HepG2^WT^ cells (Figure 6A). We next examined whether the Jun N-terminal kinase (JNK) pathway is involved in the MH-mediated cytotoxicity in HepG2^DR^ cells. Western blot analysis showed that the phosphorylation state of JNK and p38 and the expression of p53 and p53-upregulated modulator of apoptosis (PUMA) in HepG2^DR^ cells significantly increased, in a concentration-dependent manner, after CP-MH treatment compared with those in untreated HepG2^DR^ and HepG2^WT^ cells (Figure 6B).

### 2.5. MH-Mediated PARP1 Regulation Potentiates the Antitumor Effect of CP in HepG2^DR^ Hepatoma-Bearing Mice

To evaluate the antitumor effect of MH in combination with CP in an in vivo model, HepG2^DR^ cells were injected subcutaneously into the flanks of nude mice. CP-MH treatment at doses of 20 and 40 mg/kg significantly reduced the tumor volume and weight compared with those in the vehicle or CP only-treated groups (Figure 7A,B). The tumors were then resected and histologically evaluated by immunohistochemistry for PARP1 and LC3I/II expression. Reduced expression levels of PARP1 and LC3I/II were observed in tumors after CP-MH treatment, whereas the untreated HepG2^DR^ cell-xenograft tumors showed much higher PARP1 and LC3I/II expression (Figure 7C,D). The in vivo results were confirmed by immunoblotting using tumor tissue samples. The results showed a significant reduction in the expression levels of the PARP1, HMGB1, and LC3I/II proteins and a significant increase in the expression level of cleaved caspase-3 after CP-MH treatment compared with those in the vehicle-treated groups (Figure 7E). These findings suggest that combined CP-MH treatment sensitizes HepG2^DR^ cells to apoptotic cell death by regulating autophagy in vivo.

## 3. Discussion

The platinum-based drug CP has been widely used in cancer chemotherapy, including lung cancer, ovarian cancer, colon cancer, and liver cancer [12,29]. However, clinical trials showed a modest and limited efficacy of CP against HCC. Hence, it is imperative to discover a potent chemotherapeutic agent that can treat HCC, alone or in combination with conventional drugs [30]. In the present study, we demonstrate for the first time a combination effect of CP and MH on HCC HepG2^DR^ cells in vitro and in vivo. We found that intracellular PARP1 expression induces a chemo-resistance to CP in HepG2^DR^ cells. This chemo-resistance induced by PARP1 is facilitated through the cytosolic translocation of HMGB1 via PARylation, which is known to induce autophagy by disrupting the interaction between Beclin-1 and Bcl-2 [31,32]. Further, CP/MH-mediated regulation of PARP1 led to the suppression of autophagy and induction of apoptosis in HepG2^DR^ cells.

Several studies have shown that activation of autophagy by chemotherapeutic drugs is responsible, at least in part, for the development of multidrug-resistance [4,16]. Therefore, a combination of chemotherapeutic drugs with an appropriate autophagy inhibitor has been proposed as an effective approach for the reversal of the tumor cell profile from chemo-resistant to sensitive [33,34]. Although the role of CP in autophagy induction has already been reported [9], the cellular mechanisms of this drug remain to be explored. CP-induced autophagy activation suppresses the apoptosis and protects cells from cytotoxicity; in contrast, few other studies have reported the role of CP in the potentiation of autophagic cell death [35,36]. In the present study, we show that CP-induced autophagy protects HepG2^DR^ cells by activation of PARP1, however, targeting PARP1 by using CP/MH and ANI augmented the cytotoxic effect of CP. Therefore, the functional significance of autophagy activation is cellular milieu-dependent, and factors such as the cell type, drugs, and the duration of treatment determine the fate of autophagic cells for their survival or death.

PARP1-regulated autophagy appears to be mediated through the PARylation of HMGB1, a highly conserved chromatin-associated nuclear protein that plays a critical role in the regulation of autophagy and apoptosis by acting as an activator of autophagy and suppressor of apoptosis [37,38]. The functional role of HMGB1 in cancer development and progression has been reported to vary, depending upon the cell line. In addition, HMGB1 is activated in CP-resistant cancer cells [39,40]. Here, we showed that the expression, translocation, and release of HMGB1 are PARP1-dependent, which demonstrates the modulatory effect of CP on HepG2^DR^ cells via autophagy activation. Based on the possibility that PARP1 expression assists HMGB1 in nucleo-cytoplasmic shuttling via PARylation, we suggest that an increased amount of cytoplasmic HMGB1 could activate autophagy by disrupting the interaction between Beclin-1 and BCL-2. In our study, we found that CP/MH and EP treatments suppressed the cytoplasmic expression of HMGB1, which in turn inhibited autophagy and activated apoptosis in CP-resistant HepG2^DR^ cells. Based on these findings, we speculate that HMGB1-mediated autophagy activation may be involved in the development of CP-resistance in HepG2^DR^ cells.

CP treatment of HepG2^DR^ cells and xenografts induced changes in the physiological and biochemical characteristics of autophagy, such as accumulation of acidic vacuoles and increased LCI/II expression [41]. To directly determine the role of autophagy in CP-resistant HepG2^DR^ cells, we assessed the expression levels of major proapoptotic and autophagic markers after CP treatment, combined with MH, 3-MA, and RNA interference with the *ATG5* gene expression in vitro. Remarkably, both CP/MH and CP/3-MA treatments and an *ATG5* siRNA knockdown potentiated HepG2^DR^ cells to undergo apoptosis by inhibiting autophagy [9,42]. However, in vivo CP/MH treatment resulted in a marked decrease in PARP1 and LC3I/II expression, which is directly involved in the survival of CP-resistant HCC cells. Several studies demonstrate the low cytotoxicity of Morin on cellular cultures and animal models. Morin showed weak cytotoxic effects (IC50 = 250 ± 40 µM) on human leukemia cells [19]. Furthermore, animal studies showed no toxic effects of Morin on rats. After 13 weeks of treatment with high Morin doses (from about 300 to 2400 mg/Kg), the rats did not show any adverse effects. Based on these observations, the authors calculated the no-adverse-effect level of Morin at about 300 mg/Kg of body weight/day [43]. Our results show that Morin at very low concentrations (20 or 40 µM) reverse cisplatin resistance of HCC in vitro and in vivo, suggesting that the combination of Morin hydrate and cisplatin may be a promising therapeutic strategy for the efficacy of conventional chemotherapeutic drugs.

## 4. Materials and Methods

### 4.1. Chemicals

CP, MH, Dulbecco’s modified Eagle’s medium (DMEM), 4′-6-diamidino-2-phenylindole (DAPI), acridine orange (AO), Hoechst 33342, rhodamine-123, 2′,7′-dichlorofluorescein diacetate (H_2_DCFDA), JC-1 dye, 3-methyladenine (3-MA), 4-amino-1,8-naphthalimide (ANI), ethyl pyruvate (EP), and 3-(4,5-dimethylthiazol-2-yl)-2,5-diphenyltetrazolium bromide (MTT) were purchased from Sigma–Aldrich (St. Louis, MO, USA). Fetal bovine serum (FBS) was supplied by Gibco (Gaithersbug, MD, USA). All solvents were of cell culture grade and supplied by Sigma–Aldrich. A detailed list of antibodies used in this study is included in Appendix A.

### 4.2. Cell Culture

HepG2 human liver hepatoblastoma cells (ATCC) were cultured in DMEM supplemented with 10% heat-inactivated FBS and 1% penicillin/streptomycin in an atmosphere of 5% CO_2_ in air at 37 °C. To develop CP-resistant cells, wild-type HepG2 cells (HepG2^WT^) were cultured with CP (2 μM), and the surviving cells were treated with progressively increasing CP concentrations during cell passages. After several rounds of selection, a clone named HepG2^DR^ was obtained.

### 4.3. ATPase Activity Assay

Cells were incubated at 37 °C for 30 min in ATPase assay buffer. The reaction was initiated by the addition of 5 mM ATP, and incubated for 20 min at 37 °C. SDS solution (0.1 mL of 5% SDS) was added to terminate the reaction, and the amount of inorganic phosphate released was quantified colorimetrically (Sigma-Aldrich).

### 4.4. Cell Viability Assay

HepG2^DR^ cells (1 × 10^5^ cells/well) were seeded in each well of a 96-well plate, and incubated with either CP (20 μM) or MH (20 and 40 μM) for 24 h at 37 °C. The exposed cells were then treated with MTT solution (5 mg/mL) for an additional 4 h. The dark-blue formazan crystals that formed in the cells were dissolved in DMSO, and the absorbance was measured at 540 nm using an ELISA plate reader (Bio-Tek Instrument Co., WA, USA). The viable cells were calculated with respect to the percentage of control by the following formula:(Treated group/control group) × 100

### 4.5. Apoptosis Assay

As DNA in cells showing apoptotic characteristics are sensitive to formamide, denatured DNA was detected using a monoclonal antibody against single-stranded (ss) DNA using an ApoStrand™ ELISA apoptosis detection kit (Enzo Life Sciences, Plymouth Meeting, PA, USA) according to the manufacturer’s instructions.

### 4.6. Colony Formation Assay

A colony formation assay was performed to evaluate the cytotoxicity of CP and CP/MH for HepG2^DR^ cells. Briefly, cells were seeded in the complete medium in 6-well plates (BD Falcon, Sandiego, CA, USA) at a density of 300 cells per well and allowed to attach. Then, the cells were exposed to various concentrations of the corresponding drugs for 24 h. Subsequently, the cells were washed with phosphate-buffered saline (PBS) and cultured in a fresh medium until colonies were formed (10 days). The colonies were washed, fixed with methanol/acetic acid (3:1) for 10 min, and stained with 0.1% (*w*/*v*) crystal violet for 10 min. After being thoroughly washed with PBS, colonies were counted manually using the ImageJ software (www.rsbweb.nih.gov/ij/).

### 4.7. Determination of Autophagy

Acidic intracellular compartments were visualized by AO staining. After 24 h of incubation with various concentrations of CP and CP/MH, HepG2^DR^ cells were washed with PBS and stained with AO (10 μg/mL) for 15 min at 37 °C. Subsequently, the cells were washed with PBS and visualized for red fluorescence under a fluorescence microscope (Eclipse TS200; Nikon Corp., Tokyo, Japan) at 400× magnification.

To determine the specific location of autophagic vacuoles, HepG2^DR^ cells were seeded on glass coverslips (SPL, Korea) in 12-well cell-culture plates. Cells were then treated with various concentrations of the corresponding drugs for 24 h at 37 °C. After treatment, cells were fixed with 4% paraformaldehyde in PBS for 10 min. Fluorescence microscopy of cells stained with Cyto-ID Green Detection Reagent (Enzo Life Sciences, Plymouth Meeting, PA, USA) was performed according to the manufacturer’s protocol. In brief, cells were washed with PBS, stained with Cyto-ID in an indicator-free medium supplemented with 5% FBS for 30 min at 37 °C. Cells were counter stained with Hoechst-33342 to stain the nucleus. The coverslips were washed with assay buffer and visualized under a fluorescence microscope (Nikon Eclipse TS200, Nikon Corp., Tokyo, Japan) at 400× magnification.

### 4.8. Determination of Intracellular ROS

The H_2_DCFDA deacetylated in cells, where it reacts quantitatively with intracellular free radicals and get converted into its fluorescent product. To detect intracellular reactive oxygen species (ROS) production, HepG2^DR^ cells were treated with various concentrations of CP and MH, and incubated for 24 h at 37 °C. After incubation cells were rinsed with PBS, followed by addition of 10 µM peroxide-sensitive fluorescent probes H_2_DCFDA and incubated for 30 min at 37 °C. The ROS production was analyzed under a fluorescence microscope.

### 4.9. Isolation of Mitochondrial Proteins

A mitochondrial-enriched fraction was prepared from HepG2^DR^ cells grown to 90% confluence. The cells were pelleted by centrifugation and resuspended in mitochondrial extraction buffer (70 mM sucrose, 200 mM mannitol, 10 mM 4-(2-hydroxyethyl)-1-piperazineethanesulfonic acid, 1 mM ethylene glycol-*bis*(β-aminoethyl ether)-*N*,*N*,*N*′,*N*′-tetraacetic acid, pH 7.5), followed by homogenization using a Dounce homogenizer and centrifugation at 600× *g* for 10 min. The resulting pellet was discarded, and the supernatant was transferred into a new Eppendorf tube and centrifuged at a high speed (11,000× *g*, 10 min) to separate the mitochondria from soluble cytosolic proteins. The protein concentration was measured using the Bradford method. 

### 4.10. Determination of Mitochondrial Membrane Potential

Mitochondrial membrane potential was monitored with the JC-1 fluorescence dye, a cell-permeable cationic dye that preferentially partitions into mitochondria based on the highly negative membrane potential. HepG2^DR^ cells were placed on coverslips in 12-well plates and exposed to various concentrations of CP and MH for 24 h at 37 °C. The treated cells were rinsed with PBS and incubated with JC-1 (1 μg/mL) for 30 min at 37 °C. Cells were then rinsed with PBS and viewed under a fluorescence microscope at 200× magnification.

### 4.11. Immunoblot Analysis

HepG2^DR^ cells were treated with the indicated amount of CP and CP-MH for 24 h at 37 °C. After incubation, cells were harvested and lysed with RIPA lysis buffer (Sigma, St. Louis, MO, USA) and protein concentrations were quantified by Bradford assay [44]. For Western blot analysis, an equal amount of protein (30 μg in each lane) was subjected to SDS-polyacrylamide gel electrophoresis, and transferred to a polyvinyldenefluoride (PVDF) membrane (Roche Diagnostics, Indianapolis, IN, USA) by electroplating. The blots were probed with primary antibodies followed by horseradish peroxidase-conjugated secondary antibody, and visualized by enhanced chemiluminescence (ECL).

### 4.12. Detection of Nuclear DNA Condensation with Hoechst-33342

HepG2^DR^ cells were treated with the Hoechst 33342 stain according to Singh et al. Briefly, cells were cultured and treated with different concentrations of CP and MH for 24 h at 37 °C, then washed twice with PBS and fixed with cold 4% formaldehyde. The cells were then washed again with PBS and incubated with Hoechst 33342 (1 µg/mL) for 10 min at 37 °C. After washing cells with PBS, fluorescence was detected under a fluorescence microscope (Nikon Eclipse TS200) at 400× magnification.

### 4.13. Quantitative Real-Time PCR Analysis

Total RNA was isolated using the RNA-spin™ Total RNA Extraction Kit (Intron Biotechnology, Korea), according to the manufacturer’s protocol. Concentrations of RNA were analyzed using a Qubit22^®^ 2.0 fluorimeter RNA Assay Kit (Life technologies, USA). cDNA synthesis was carried out using a Maxime RT Premix cDNA Synthesis Kit (Intron Biotechnology, Korea), according to the manufacturer’s protocol. The RT-PCR control (β-actin) was amplified under the same PCR parameters and used to normalize the quantitative data. Quantitative real-time PCR was performed using the Agilent technology qPCR System (CA, USA). PCR primers used for PARP1, LC3, Beclin-1, Cyt *c,* and Casp-3 are mentioned in Appendix A.

### 4.14. Immunostaining

Cells were cultured on a cell culture cover glass (SPL, Republic of Korea) for 24 h before treatment. After treatment with CP (20 μM) and CP (20 μM)-MH (20 and 40 μM), cells were fixed with 4% paraformaldehyde in phosphate-buffered saline (PBS) for 20 min, blocked with 3% normal goat serum, and incubated overnight at 4 °C with HMGB1 primary antibody. Cells were then incubated with FITC labeled goat anti-rabbit IgG secondary antibody for 1 h. The coverslips were washed with PBS, mounted on glass slides with DAPI anti-fade solution (Invitrogen), and visualized under an EPI fluorescence microscope at 200× magnification.

### 4.15. siRNA Transfection

HepG2^DR^ cells were seeded in a 6-well plate and transfected with 25 pM siRNA against Atg5 (Atg5 no.s16537, Ambion, Life Technologies, Darmstadt, Germany) using 7.5 μL Lipofectamine^®^ RNAiMAX Transfection Reagent (Invitrogen) [45]. Cells were lysed after 24 h of CP (20 μM), and CP (20 μM)-MH (20 and 40 μM) exposure, and expression levels of Atg5 and β-actin were observed by Western blot analysis. siRNA sequences used for Atg5 are mentioned in Appendix A.

### 4.16. Animals and Establishment of a Mouse Xenograft Model

Four-week-old male BALB/c-nu-nude mice (15 ± 5 g) were obtained from Oriental Biotechnology (Korea). The animals were maintained under specific pathogen-free conditions, with food and water supplied ad libitum, at the Animal Care Laboratory of Daegu University (LMO No. LML-16-1134). All animal experiments were carried out in accordance with the Guide for the Care and Use of Laboratory Animals of the National Institutes of Health and were approved by the Committee on the Ethics of Animal Experiments of Daegu University, South Korea (DUIACC-2017-1-0218-001). A total of 5 × 10^6^ cells were subcutaneously injected into the right flank of the mice. Tumor volume was calculated using the following formula: V = (L × W^2^)/2, where L is the length and W is the width of the tumor nodules measured with Vernier calipers. Once the volume of the tumors reached 25–50 mm^3^, the mice were randomly divided into four groups (*n* = 5). The mice were treated twice a week for 4 weeks with IV injections of either vehicle (0.1% DMSO), CP (2 mg/kg body weight), and/or MH (20 and 40 mg/kg body weight). Body weights were measured before each dose. After the twentieth day of dosing, the mice were euthanized and the tumors were isolated, weighed, and photographed. Immunoblotting and histological analysis were performed on the hepatoma xenografts. Briefly, the samples were fixed in a 10% formalin solution, processed, embedded in paraffin, sectioned, and IHC stained using antibodies against PARP1 and LC3I/II. The IHC images were obtained under the light microscope at 200× magnification. The immunoreactive areas in the IHC images were quantified using ImageJ software.

### 4.17. Statistical Analysis

Statistical analysis was performed with SPSS (Statistical Package for the Social Sciences) 22.0. Results are expressed as the mean ± standard deviation (SD) of three independent experiments. The data were subjected to one-way analysis of variance (ANOVA) and the significance of difference between sample means was calculated by Duncan’s multiple range test (SPSS Inc., Chicago, IL, USA). Differences were considered statistically significant for values of *p* < 0.05.

## 5. Conclusions

These findings indicate that PARP1 hyper-activation promotes autophagy under CP therapy stress, leading to HepG2^DR^ cell survival both in vitro and in vivo. The MH inhibits CP-induced autophagy through suppression of PARP1 activation, which leads to HepG2^DR^ cell death via mitochondrial apoptosis. Based on these findings, we propose the use of a CP/MH combination as a promising therapeutic strategy to enhance the efficacy of conventional chemotherapeutic drugs, thereby improving clinical outcomes in HCC patients by targeting the PARP1/HMGB1 pathway of autophagy.

## Figures and Tables

**Figure 1 cancers-11-00986-f001:**
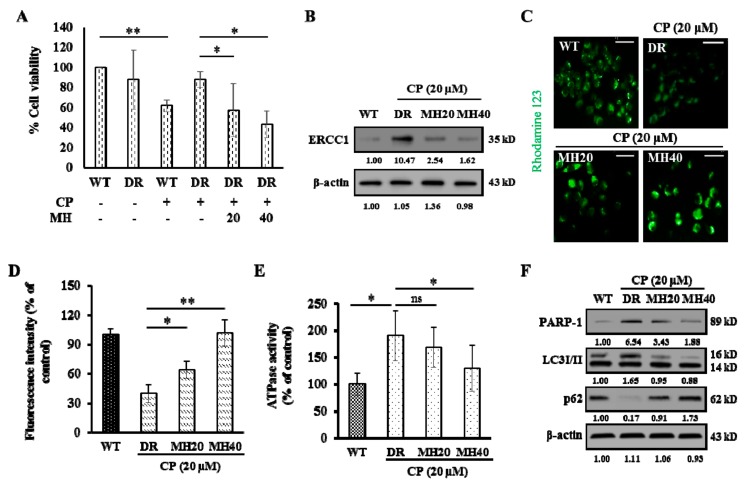
Morin hydrate (MH) reverses cisplatin (CP) resistance of HepG2^DR^ cells. (**A**) Cell viability of HepG2^WT^and HepG2^DR^ cells was measured after CP and CP-MH treatment by the MTT assay. (**B**) Western blot analysis of excision repair cross-complementation group 1 (ERCC1) expression in HepG2^DR^ cells as a marker of developing CP resistance; ERCC1 levels were normalized to those of β-actin. (**C**) Fluorescent micrograph of rhodamine-123 intracellular accumulation in HepG2^DR^ cells using an ATP-binding cassette sub-family B member 1 (ABCB1 or P-gp) activity assay after 24 h of corresponding drug treatment (magnification: 200×; scale bar: 0.1 mm). (**D**) Fluorescence intensity calculated using the ImageJ software. (**E**) ATPase activity of HepG2^DR^ cells after 24 h of CP-MH treatment. (**F**) Expression levels of poly(ADP-ribose) polymerase 1 (PARP1), LC3I/II, and p62 in HepG2^DR^ cells were analyzed by Western blotting; results were normalized relative to β-actin. Relative intensities were measured by ImageJ software. Data are presented as the means ± standard deviation (SD) (*n* = 3). * *p* < 0.05, ** *p* < 0.01. DR = drug resistant; WT = wild type.

**Figure 2 cancers-11-00986-f002:**
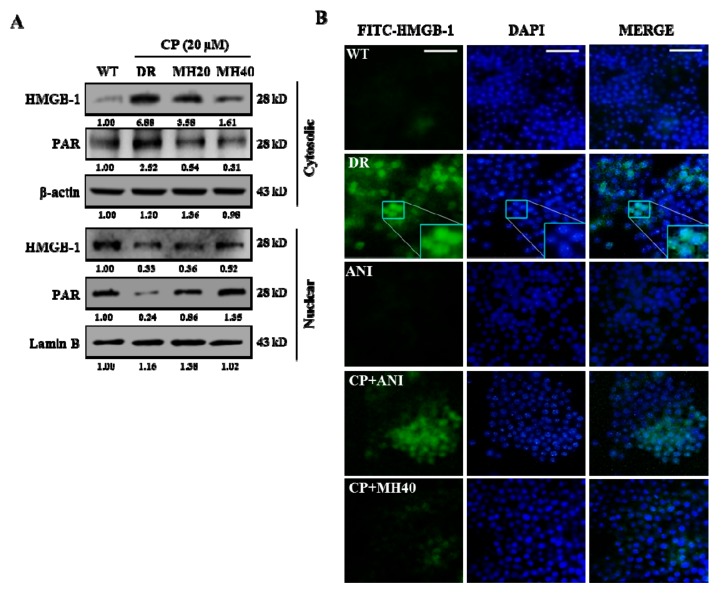
MH suppresses CP-induced cytoplasmic translocation of high-mobility group box 1 (HMGB1). (**A**) Expression levels of HMGB1 and PAR in cytosolic and nuclear fractions of HepG2^DR^ cells were measured by Western blotting after CP-MH treatments; results were normalized relative to β-actin and lamin B, respectively. Relative intensities were measured by ImageJ software. At least two independent experiments were performed. (**B**) Immunocytofluorescence of HepG2^WT^ and HepG2^DR^ cells for cytosolic HMGB1 translocation after ANI, CP-ANI, and CP-MH treatments, performed using a fluorescein isothiocyanate (FITC)-labeled anti-HMGB1 antibody (magnification: 400×; scale bar: 0.1 mm); DAPI (blue) was used as a nuclear stain.

**Figure 3 cancers-11-00986-f003:**
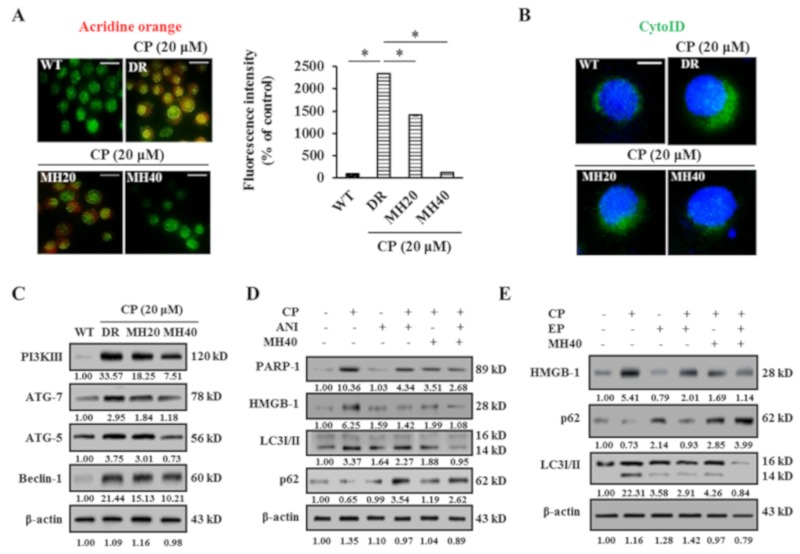
MH suppresses PARP1-mediated autophagy in HepG2^DR^ cells. (**A**) HepG2^WT^ and HepG2^DR^ cells were stained with the acridine orange (AO) dye after corresponding drug treatments and observed under a fluorescence microscope. Autophagosomes are indicated by red fluorescence (magnification: 400×; scale bar: 0.1 mm). Autophagy rate (% of control) was calculated using the ImageJ software (left panel). (**B**) Fluorescence images of HepG2^DR^ cells, showing autophagic acidic vesicles, were obtained by staining with the Cyto-ID^®^ autophagy reagent (green) after CP-MH treatments; the nucleus is stained with Hoechst 33342 (blue) (scale bar: 0.1 mm). (**C**) Expression levels of major autophagy-associated markers were analyzed by Western blotting using specific antibodies against PI3KIII, ATG7, ATG5, and Beclin-1 after treatment with CP-MH; results were normalized relative to β-actin. (**D**) Expression levels of cytosolic PARP1, HMGB1, LC3I/II, and p62 after treatment with CP, MH, and ANI, alone or in combination, were analyzed by Western blotting. Results were normalized relative to β-actin. (**E**) Expression levels of cytosolic HMGB1, p62, and LC3I/II were evaluated after treatment with CP, MH, and EP, alone or in combination. Results were normalized relative to β-actin. Relative intensities were measured by ImageJ software. At least two independent experiments were performed. * *p* < 0.05, ** *p* < 0.01.

**Figure 4 cancers-11-00986-f004:**
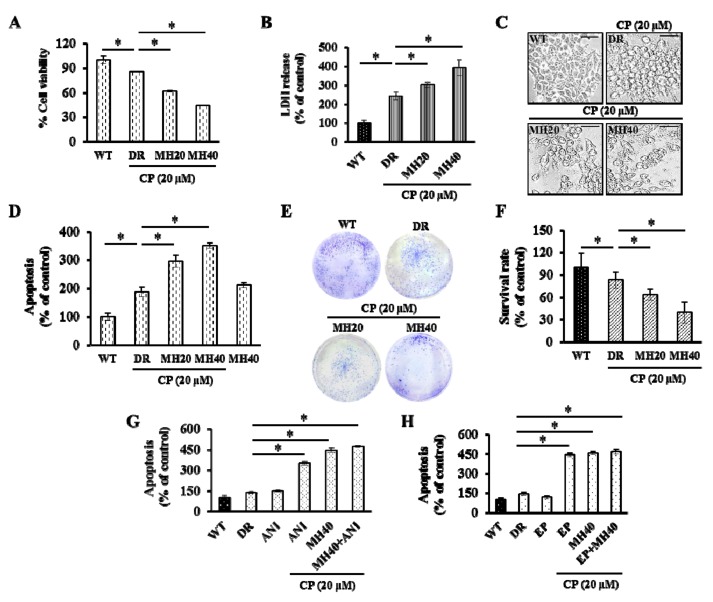
MH promotes apoptosis of HepG2^DR^ cells against CP. (**A**) HepG2^DR^ cell viability was measured after drug treatment by the MTT assay. (**B**) Total cellular LDH release was measured by the LDH assay. (**C**) Representative microscopic images of the morphology of HepG2^WT^ and HepG2^DR^ cells after drug treatments (magnification: 200×; scale bar: 0.1 mm). (**D**) Percentage of apoptotic HepG2^DR^ cells was determined by the apoptosis assay. (**E**) HepG2^DR^ cell proliferation was evaluated by the colony formation assay. (**F**) The colony survival rate was calculated using the ImageJ software. (**G**,**H**) The apoptosis assay with HepG2^DR^ cells was performed after 24 h of treatment with the indicated concentrations of CP in the presence or absence of ANI (10 μM), ethyl pyruvate (EP) (20 mM), and MH (40 μM), using the ApoStrand™ ELISA apoptosis detection kit. Data are presented as the means ± SD (*n* = 3). * *p* < 0.05, ** *p* < 0.01.

**Figure 5 cancers-11-00986-f005:**
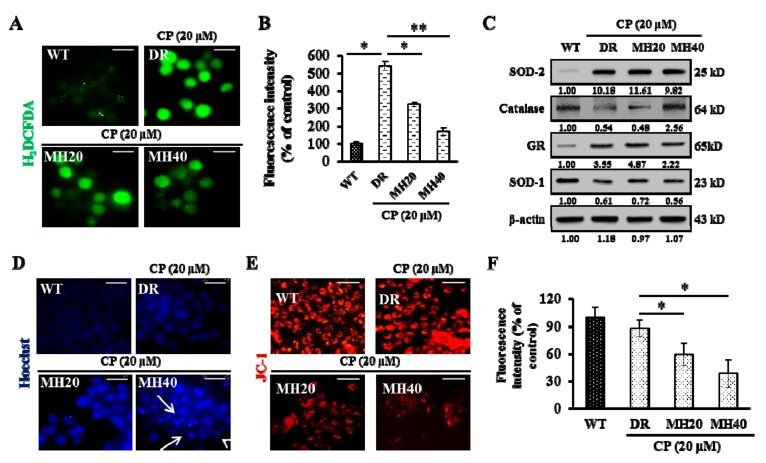
Effect of MH on CP-induced oxidative stress in HepG2^DR^ cells (**A**) Intracellular reactive oxygen species (ROS) production in HepG2^WT^ and HepG2^DR^ cells after drug treatments was observed under a fluorescence microscope (magnification: 400×; scale bar: 0.1 mm). (**B**) Fluorescence intensity was measured using the ImageJ software. (**C**) Western blot analysis of antioxidant markers was performed using specific antibodies against SOD-2, catalase, GR, and SOD-1; β-actin was used as a loading control. Densitometry analysis was carried out using the ImageJ software. At least two independent experiments were performed. (**D**) HepG2^WT^ and HepG2^DR^ cells were stained with Hoechst 33342 after drug treatments and assayed under a fluorescence microscope (magnification: 400×; scale: 0.1 mm). White arrows indicate bright blue pyknotic nuclei. (**E**) Mitochondrial membrane potential disruption in HepG2^WT^ and HepG2^DR^ cells was assayed using the red JC-1 stain after drug treatments and visualized by fluorescence microscopy (magnification: 200×; scale bar: 0.1 mm). (**F**) Fluorescence intensity of the JC-1 red stain was measured using the ImageJ software. Data are presented as the mean ± SD (*n* = 3). * *p* < 0.05, ** *p* < 0.01.

**Figure 6 cancers-11-00986-f006:**
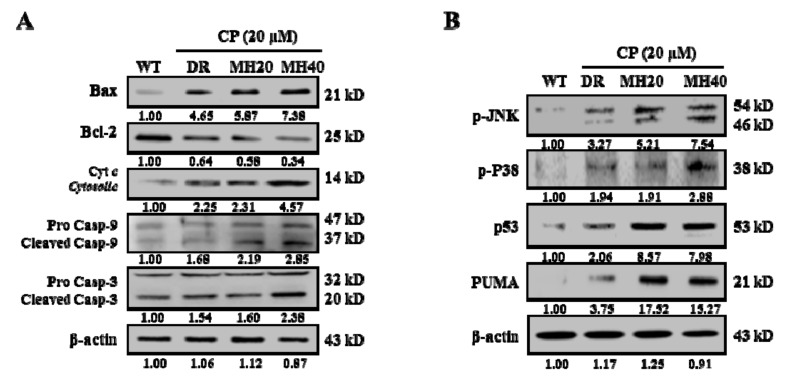
MH regulated the ratio of Bcl-2/Bax and the activation of p38/JNK stress activated protein kinase. (**A**) Expression levels of major pro-apoptotic and anti-apoptotic markers (Bax, Bcl-2, Cyt *c,* procaspase-9/cleaved caspase-9, and procaspase-3/cleaved caspase-3) were measured in cytosolic and mitochondrial fractions of HepG2^WT^ and HepG2^DR^ cells by Western blotting; β-actin was used as a loading control. (**B**) Expression levels of major apoptosis signaling markers (p-JNK, p38, p-p38, p53, and p53-upregulated modulator of apoptosis (PUMA)) were evaluated in HepG2^WT^ and HepG2^DR^ cells by Western blotting; results were normalized relative to β-actin. Relative intensities were measured by ImageJ software. At least two independent experiments were performed.

**Figure 7 cancers-11-00986-f007:**
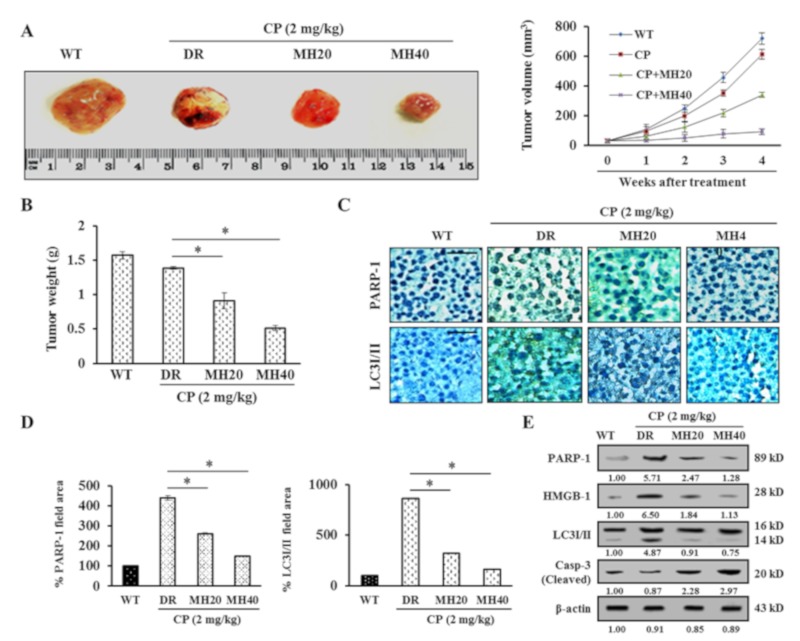
MH potentiates the antitumor effect of CP on HepG2^DR^ cell xenograft. (**A**) Representative images showing regression in tumor volumes after treatment with CP-MH compared with those in the vehicle-treated groups of HepG2^DR^ cell xenografts in nude mice. (**B**) Surgically resected tumors from vehicle and CP-MH-treated xenograft mice; changes in tumor weights were observed after initiation of treatment. (**C**) Expression levels of PARP1 and LC3I/II were evaluated using an immunohistochemistry assay with specific antibodies (magnification: 200×; scale bar: 0.1 mm). (**D**) Percentage of the area of representative proteins (PARP1 and LC3I/II) was quantified using the ImageJ software. (**E**) Expression level analysis of PARP1, HMGB1, LC3I/II, and cleaved caspase-3 by Western blotting after CP-MH treatments in HepG2^DR^ cell xenograft tissues. Relative intensities were measured by ImageJ software. Data represent the mean ± SD of three independent experiments (*n* = 5). * *p* < 0.05, ** *p* < 0.01.

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
