# Peer review of "Morin Hydrate Reverses Cisplatin Resistance by Impairing PARP1/HMGB1-Dependent Autophagy in Hepatocellular Carcinoma"

_cancers, 2019, doi:10.3390/cancers11070986_

Round 1
Reviewer 1 Report
These are the few things authors need to address
The authors have demonstrated their results in only one cell line, cancer is a heterogenous disease and it's ideal to demonstrate the authors findings in at least 2-3 HCC cell lines
PARP effects chemoresistance of many DNA damaging agents can authors comment on it's role in other DNA damaging drugs used in treat of HCC in first line or second line treatment
Authors throughout manuscript mention significant difference, but do not mention p value in text or figure, it's confusing to see what they are talking about. Further, std. deviation are high in many cases (example Figure 1A and 1E), not sure if there is significant difference in what are saying.
Some western blots and confocal images need to be quantified (westerns normalizing to loading control ) and confocal images by number of cells. It's hard to interpret the data by images
For any cancer drugs or a chemical entering into human body, one needs to perform PK/PD studies and determine what is the maximum tolerated dose determined by steady state plasma level concentration, taking into effect like protein binding ability and half-life. Author have used Morin hydrate, and showed it inhibits the HCC cancer cell line. What is the dose which can be achievable in humans? Is the dose used in cell lines correlate to humans? Authors should make a detailed discussion about this in the paper.
Mitochondrial depolarization (MMD) leads to activation of pro-apoptotic proteins leads to activation of caspases , which leads to apoptosis. Authors have shown increase of pro-apoptotic proteins, caspases and apoptosis but decrease of MMD with MH treatment, this result is confusing and not in line with others.
Author Response
Point 1. The authors have demonstrated their results in only one cell line, cancer is a heterogenous disease and it's ideal to demonstrate the authors findings in at least 2-3 HCC cell lines
Response 1: Thank you for your valuable suggestion. Although we have already studied the role of Morin hydrate in different hepatocellular carcinoma (HCC) cell lines like Hep3B, Huh7 and SK-Hep1; the anti-tumor effect of Morin hydrate was evaluated significantly in all these cell lines by observing the dramatic change in cell viability and autophagy activation including HepG2 cells in combination of CP. The following results of the present study are in preparation for next publication. In the meantime, we are also in progress to develop cisplatin and 5-FU drug resistant Hep3B, Huh7 and SK-Hep1 cells in our laboratory, hopefully we will come up with novel findings in our next reports.
Point 2. Authors throughout manuscript mention significant difference, but do not mention p value in text or figure, it's confusing to see what they are talking about. Further, std. deviation are high in many cases (example Figure 1A and 1E), not sure if there is significant difference in what are saying.
Response 2: We thank the reviewer for making us aware our error. We have now mark symbol of statistical analysis in all figure graphs.
Point 3. Some western blots and confocal images need to be quantified (westerns normalizing to loading control) and confocal images by number of cells. It's hard to interpret the data by images.
Response 3: We appreciate the reviewer’s remarks. We have now quantified the band intensities in Fig. 1B, 1F, 2A, 3C, 3D, 3E, 5C, 6A, 6C, and 7E using image J software.
Point 4. For any cancer drugs or a chemical entering into human body, one needs to perform PK/PD studies and determine what is the maximum tolerated dose determined by steady state plasma level concentration, taking into effect like protein binding ability and half-life. Author have used Morin hydrate, and showed it inhibits the HCC cancer cell line. What is the dose which can be achievable in humans? Is the dose used in cell lines correlate to humans? Authors should make a detailed discussion about this in the paper.
Response 4: The reviewer makes an excellent point. Several studies demonstrate the low cytotoxicity of Morin on cellular cultures and animal models. Morin showed weak cytotoxic effects (IC50=250 ± 40 uM) on human leukemia cells (Caselli et al. Curr Med Chem. 2016, 23, 774-91). Furthermore, animal studies showed no toxic effects of Morin on Rats. After 13 weeks treatment with high Morin doses (from about 300 to 2400 mg/Kg), the rats did not show any adverse effects. Based on these observations, authors calculated the no-adverese-effect level of Morin at about 300 mg/Kg of body weight/day. Our results show that Morin at very low concentration (20 or 40 uM) reverse cisplatin resistance of HCC in vitro and in vivo, suggesting that the combination of Morin hydrate and cisplatin may be a promising therapeutic strategy to the efficacy of conventional chemotherapeutic drugs.
Now we have addressed this in discussion section.
Point 5. Mitochondrial depolarization (MMD) leads to activation of pro-apoptotic proteins leads to activation of caspases, which leads to apoptosis. Authors have shown increase of pro-apoptotic proteins, caspases and apoptosis but decrease of MMD with MH treatment, this result is confusing and not in line with others.
Response 5: JC-1 is a mitochondrial dye that stains mitochondria in living cells in a membrane potential dependent manner. In Fig. 5F and 5G, a significant loss in the red fluorescence intensity of JC-1 is observed in HepG2DR cells treated with cisplatin and MH, indicating that co-treatment of MH with cisplatin promotes loss of mitochondrial membrane potential, release of cytochrome-c, and thus activate apoptotic caspase. Now we modified the sentence in the result section for better clarity
Reviewer 2 Report
The experimental design and results of this paper are very organized. Author found morin hydrate could reverses cisplatin resistance that is very valuable. However, I have a little comments:
In this paper, how to sampling the fluorescent stain image to enter the Image analysis, the experimental method did not explain, that only showed using image software analysis.
All figures did not exhibit symbols of statistical analysis. Their statistical symbols should be added.
In the legend of Fig. 7A (P10), the symbol (Mh) is different from the others (MH). In addition, it did not indicate whether it was used together with the CP? Even, it was not explained in the experimental methods (P13, L513).
Author Response
The experimental design and results of this paper are very organized. Author found morin hydrate could reverses cisplatin resistance that is very valuable. However, I have a little comments:
Point 1. In this paper, how to sampling the fluorescent stain image to enter the Image analysis, the experimental method did not explain, that only showed using image software analysis.
Response 1: Cells were cultured on a cell culture cover glass (SPL, Republic of Korea) for 24 h before treatment. After treatment with CP (20 μM), and CP (20 μM)-MH (20 and 40 μM), cells were fixed with 4% paraformaldehyde in phosphate-buffered saline (PBS) for 20 min, blocked with 3% normal goat serum, and incubated overnight at 4°C with HMGB1 primary antibody. Cells were then incubated with FITC labeled goat anti-rabbit IgG secondary antibody for 1 h. The coverslips were washed with PBS, mounted on glass slides with DAPI anti-fade solution (Invitrogen, USA), and visualized under an EPI fluorescence microscope at 200× magnification.
Now we explain this in materials and methods section
Point 2. All figures did not exhibit symbols of statistical analysis. Their statistical symbols should be added.
Response 2: We thank the reviewer for making us aware our error. We have now mark symbol of statistical analysis in all figure graphs.
Point 3. In the legend of Fig. 7A (P10), the symbol (Mh) is different from the others (MH). In addition, it did not indicate whether it was used together with the CP? Even, it was not explained in the experimental methods (P13, L513).
Response 3: We appreciate the reviewer for correcting our errors. We made the correction.
Round 2
Reviewer 1 Report
NA